# Prevalence of SARS-CoV-2 IgG/IgM Antibodies among Danish and Swedish Falck Emergency and Non-Emergency Healthcare Workers

**DOI:** 10.3390/ijerph18030923

**Published:** 2021-01-21

**Authors:** Jannie Laursen, Janne Petersen, Maria Didriksen, Kasper Iversen, Henrik Ullum

**Affiliations:** 1Department of Global Business Quality Management, Falck, Sydhavnsgade 18, 2450 Copenhagen, Denmark; 2Center for Clinical Research and Prevention, Copenhagen University Hospital, Bispebjerg and Frederiksberg, Bispebjerg Bakke 23, 2400 Copenhagen, Denmark; janne.petersen.01@regionh.dk; 3Section of Biostatistics, Department of Public Health, University of Copenhagen, Øster Farimagsgade 5, 1353 Copenhagen, Denmark; 4Department of Clinical Immunology, Copenhagen University Hospital, Rigshospitalet, Blegdamsvej 9, 2100 Copenhagen, Denmark; heul@ssi.dk; 5Department of Emergency Medicine and Department of Cardiology, Copenhagen University Hospital, Herlev-Gentofte, Borgmester Ib Juuls Vej 1, 2730 Herlev, Denmark; Kasper.Karmark.Iversen@regionh.dk

**Keywords:** COVID-19, epidemiology, communicable/infectious diseases, employee health, healthcare worker/homecare worker

## Abstract

Background: Knowledge about the COVID-19 outbreak is still sparse, especially in a cross-national setting. COVID-19 is caused by a SARS-CoV-2 infection. The aim of the study is to contribute to the surveillance of the pandemic by bringing new knowledge about SARS-CoV-2 seropositivity among healthcare workers. It seeks to evaluate whether certain job functions are associated with a higher risk of being infected and to clarify if such association is mediated by the number of individuals that employees meet during a workday. In addition, we investigate regional and national differences in seroprevalence. Methods: This research involved a bi-national prospective observational cohort study including 3272 adults employed at Falck in Sweden and Denmark. Participants were tested for SARS-CoV-2 antibodies every second week for a period of 8 weeks from 22 June 2020 until 10 August 2020. Descriptive statistics as well as multivariable logistic regression analyses were applied. Results: Of the 3272 Falck employees participating in this study, 159 (4.9%) tested positive for SARS-CoV-2 antibodies. The seroprevalence was lower among Danish Falck employees than among those from Sweden (2.8% in Denmark and 8.3% in Sweden). We also found that the number of customer or patient contacts during a workday was the most prominent predictor for seropositivity and that ambulance staff was the most vulnerable staff group. Conclusion: Our study presents geographical variations in seroprevalence within the Falck organization and shows evidence that social interaction is one of the biggest risk factors for becoming infected with SARS-CoV-2.

## 1. Introduction

In December 2019, the first case of COVID-19 caused by severe acute respiratory syndrome coronavirus 2 (SARS-CoV-2) was identified. On 30 January 2020, the World Health Organization (WHO) declared the SARS-CoV-2 outbreak to be a public health emergency of international concern [1]. Later it evolved into a pandemic, and to date, there have been more than 70 million confirmed cases worldwide, of whom 1.6 million have died while infected with the virus [2]. Several strategies for preventing the spread of the virus have been implemented across the world. A strong strategy, which has been widely used, is social distancing. However, for some job functions in the health care sector this strategy is not feasible to apply. Studies from Denmark and Italy have found a higher seroprevalence among healthcare workers than among the general population [3,4,5]. Healthcare workers therefore have a particularly high risk of COVID-19. Falck is a rescue corps employing more than 30,000 healthcare workers worldwide. Most individuals employed by Falck have job functions that put them at risk of being infected with SARS-CoV-2 through interactions with customers or patients. Because such interactions vary between job functions, it is possible that the risk of infection does too. Ambulance staff are presumed to be at high risk of being exposed to individuals infected with SARS-CoV-2, whereas health professionals at clinics mainly interact with the same clients and are therefore not subjected to new people on daily basis. This staff group may therefore have a smaller risk of infection. Moreover, office workers who do not come in contact with either customers or patients may have the lowest risk. Falck also employs part-time firefighters with variable job functions, making it difficult to speculate on the potential level of exposure to SARS-CoV-2–infected individuals. To protect patients, employees and family members of employees from becoming infected with SARS-CoV-2, Falck has taken several preventive measures. These varied between Denmark and Sweden, as Falck adhered with governmental guidelines. One measure taken was hosting telephone or video consultations with patients whenever possible. Moreover, if patients or employees experienced any symptoms potentially related to COVID-19, such as fever, coughing, sore throat, headache or sore muscles, they were not allowed to come to a clinic. In addition to this, Falck encouraged all patients to focus on their hygiene and scheduled extra time in between patients for the employees to disinfect. In Denmark, ambulance staff were asked to wear masks if they suspected a patient to have COVID-19, while Swedish ambulance staff were asked to wear masks for all patient contacts, and if they suspected a patient to be infected, they were asked to wear complete protective equipment. Falck has 8000 employees in Denmark and 2000 in Sweden. For the present study we tested 2024 Falck employees in Denmark and 1248 in Sweden for SARS-CoV-2 antibodies every other week across a period of two months. The aim of the study was to contribute to the surveillance of the pandemic and to bring new knowledge about SARS-CoV-2 seropositivity among healthcare workers by evaluating whether Falck employees with certain job functions have a higher risk of being infected and clarifying if such association is mediated exclusively by the number of individuals that the employees meet during a workday. Finally, the study investigated regional and national differences in seroprevalence.

## 2. Materials and Methods

This is a bi-national prospective observational cohort study including 3272 individuals 18 years and older. Participants were included on account of being Falck employees. All Falck employees in Denmark (*n* = 8000) and Sweden (*n* = 2000) were asked to participate in the study, and 25.3% of Danish employees agreed, while 62.4% of Swedish employees did. Participants were tested for SARS-CoV-2 antibodies using the same test every second week for a period of 8 weeks from 22 June 2020 to 10 August 2020.

### 2.1. Assessing Seroprevalence

IgG and IgM antibodies against SARS-CoV-2 were measured in whole blood using the Livzon lateral flow test (Livzon Diagnostics, Zhuhai, Guangdong, China), which we validated with a specificity of 99.54% (95% CI: 98.7–99.9) and a sensitivity of 82.58% (95% CI: 75.7–88.2) [3]. The tests were done by the participants themselves, for which they received instructions in writing and on video. The test required that the participants’ put in one drop of blood and two drops of buffer (isotonic saline) in two separate cassettes. After 15 min, a conclusive test showed a control line, and if IgG or IgM were detected a separate test line for each appeared.

To ensure complete identification of participants with antibodies, test results of participants with only one antibody (IgG or IgM) detected were validated using a different brand of test. For this follow-up test, the anti-SARS-CoV-2 (IgG/IgM) POC-test (lateral flow) WONDFO was used. This test was validated in our laboratory at the Department of Clinical Immunology, Copenhagen University Hospital, Denmark. It showed a sensitivity of 94.7% (95% CI: 89.8–97.7) and a specificity of 98.7% (95% CI: 97.4–99.4) (unpublished data). The WONDFO test was done by the participants themselves in the same way as the Livzon test. A positive test was classified as a test indicating the presence of either IgG, IgM or both antibodies.

### 2.2. Covariates

On the four occasions that participants were tested for SARS-CoV-2 antibodies, they also answered a brief questionnaire. In this, they reported their job function: ambulance staff, firefighter, healthcare staff, office staff, roadside assistance or field staff. Participants were also asked to report their national region of residence during the study period and whether they had an antibody test done outside of this study. Moreover, participants indicated how many individuals they had encountered during workdays, on average, for the two weeks prior to testing: 0, 1–5, 6–10, 11–20, and more than 20.

### 2.3. Statistics

Statistical analyses were performed using Enterprise Guide 7.1. Descriptive statistics were conducted to investigate the distribution of study participants, which was presented as frequencies and percentages. National differences in the distribution were examined using *X*^2^-tests for categorial variables. The distribution of participants with a positive test was described using frequencies and percentages. Finally, multivariable logistic regression models were applied to assess the association between job function, number of person contacts and the risk of testing positive with SARS-CoV-2 antibodies.

### 2.4. Ethical Statement

Invitations to participate in the study were sent out to all Falck employees in Denmark and Sweden individually through the governmental, personal, password-protected email-system, E-boks in Denmark and Webropol in Sweden. To ensure that participants were properly informed before they consented to participate in the study, online live webinars informing them about the study were performed for Danish participants. During the webinars, participants had the possibility of asking questions directly to the study organizers. There was no requirement from the Scientific Ethics Committee of Sweden that webinars be done. Instead, Swedish participants received written information, and they were given contact information for the study organizer, which they were told to use if they had any questions. The study adheres with the General Data Protection Regulation [6] and the ‘Scientific Ethical Treatment of Health Science Research Projects’ law [7].

Moreover, participants were pseudonymized, and data handling and analysis was conducted by an external statistician with no conflicting interests. Finally, to ensure that participants did not feel pressured into taking part in the study, testing for antibodies was also offered to those employees who did not wish to participate in the study. The study was approved by the Scientific Ethics Committees of Denmark and Sweden (Denmark: Protocol number: H-20031227; Sweden: 2020-02862).

## 3. Results

In total, 3272 individuals (2024 from Denmark and 1248 from Sweden) participated in the study. Of these, 64% (*n* = 2080) participated in all four rounds of testing, 86% (*n* = 2800) participated in at least three rounds and 95% (*n* = 3096) participated in at least two rounds.

### 3.1. Characteristics of the Study Population

There were national differences in the distribution of characteristics between Danish and Swedish participants (Table 1). More men than women participated in Denmark (75.5% men), while the opposite was the case among Swedish participants (32.9% men). In both countries, about half of the participants were between the ages of 40 and 60. A smaller proportion of Danish participants reported more than ten person contacts per day (9.7%) compared to Swedish participants (20.3%).

This may be a reflection of the smaller proportion of Danish participants being employed as healthcare staff (6.2% in Denmark versus 45.4% in Sweden), while a higher proportion was employed as firefighters (26% in Denmark versus 2.2% in Sweden). Finally, the distribution of employees across the different national regions were in keeping with the size of the regions (Table 1).

### 3.2. Proportion of Positive Immune Test

Among the 3272 included employees, 29 (0.9%) did not have a valid test result. Because of the lower sensitivity compared to specificity, a person was considered positive if they had a least one positive test. After the first test, 3.3% (*n* = 107) tested positive; after the second test, 4.1% (*n* = 133) tested positive; after the third test, 4.7% (*n* = 153) tested positive; after the fourth test 4.9% (*n* = 159) tested positive for SARS-CoV-2 antibodies, corresponding to 2.8% of Danish participants and 8.3% of Swedish participants (Table 2). The group of participants aged 60 years and above had the lowest proportion of seropositivity. Ambulance staff had the highest proportion in both countries, whereas firefighters had the lowest. Among Swedish participants the proportion of seropositivity seemed to increase with the number of people contacts during a workday. A similar phenomenon was not observed among Danish participants (Table 2).

The multiple logistic regression analyses identified testing positive for SARS-CoV-2 antibodies with an increasing number of average people contacts during a workday. The estimates did not change when adjusted for age and sex; however, when region was included as a covariate, the odds ratios (ORs) attenuated, but remained statistically significant. In the full model including information on sex, age, region and type of employment, it appears that the risk of seropositivity was doubled in employees with 11–20 contacts per day compared to zero contacts per day (OR = 2.3, 95% CI: 1.2–4.6), while the risk was tripled in employees with more than 20 contacts per day (OR = 2.9, 95% CI: 1.5–5.8) (Table 3). Compared to office staff/field staff, the crude model showed that ambulance staff had the highest risk of a positive test response (OR = 2.2, 95% CI: 1.4–3.4). Healthcare staff also had an increased risk compared to office staff/field staff (OR = 1.3, 95% CI: 0.8–2.1), while firefighters and roadside assistance had a lower risk of infection (OR = 0.4, 95% CI: 0.2–0.9 and OR = 0.9, 95% CI: 0.5–1.9, respectively). When adjusted for age, sex and region, the ORs attenuated. Applying the full model, which also included number of people contacts a day, ORs attenuated further and became statistically insignificant (Table 3). Because of the low seroprevalence among Danish participants, follow-up analyses were only conducted for Swedish participants. This showed similar results (Table 4).

## 4. Discussion

Of the 3272 Falck employees participating in this study, 159 (4.9%) tested positive for SARS-CoV-2 antibodies. The seroprevalence was lower among Danish Falck employees than among those from Sweden (2.8% in Denmark and 8.3% in Sweden). We also found that the number of customer or patient interactions during a workday was the most prominent predictor for seropositivity. It is plausible that the national variance in seroprevalence between the two countries was a result of different governmental strategies for dealing with the pandemic. The seroprevalences of 2.8% and 8.3% observed among Danish and Swedish Falck employees, respectively, are higher than those observed among Danish (1.7%) [8] and Swedish (6.8%) [9] otherwise healthy blood donors. Blood donors represent an age and sex distribution similar to that of the background population between the ages 18 and 65. A partial explanation for the increased seroprevalence seen in this study is that the prevalence in the general population is expected to increase with time.

Before the study commenced, we hypothesized that employees could be divided into risk groups dependent on the suspected number of people contacts during a workday, with a higher number indicating a higher risk. The present results validate this, as analyses revealed an increasing risk of infection with an increasing number of customer or patient interactions. Furthermore, analyses showed that customer or patient interactions had a higher impact on the risk than job function did. We also found that ambulance staff had the highest risk of seropositivity. The OR attenuated when adjusted for sex, age, and national region. The OR attenuated even further and became statistically insignificant when the number of people contacts a day was added as a covariate in the statistical model. Therefore, it is plausible that the observed increased risk is explained by a related high level of customer or patient interaction, but it may also be that work exposure for ambulance staff confer a particular risk as ambulance staff cannot reject patients with possible symptoms of COVID-19. Even though Danish ambulance staff were only asked to wear masks when they came in contact with a potential COVID-19 sufferer, while Swedish ambulance staff were asked to wear masks for all patient contacts, we found a significantly higher seroprevalence among Swedish ambulance staff (14.7%) than among Danish ambulance staff (4.1%). This may be evidence that the masks carried by staff and other protective measures did not fully protect against infection acquired at work. A way to potentially increase the efficacy of protective equipment is to ask patients to wear masks as well. In support of our findings, a recently published study including healthcare workers from the capital region of Denmark found that paramedics had the highest seroprevalence out of all hospital staff (4.95% of paramedics tested positive versus 4.04% of all hospital staff) [3]. In the same study, it was also found that the seroprevalence was significantly higher in men compared to women [3], which the present study does not support. This may be explained by the fact that there are more women working as ambulance staff in Sweden, while there are more men working as such in Denmark. Moreover, the present study shows that a smaller proportion of employees above 60 years old tested positive for antibodies. This may be explained by the elderly being aware of their increased risk of more severe COVID-19 illness and therefore taking more personal precautions to avoid infection. A specific strength of the present study is the inclusion of participants from different countries and from different regions within these countries. It is also a strength that the participants were tested for antibodies several times across the study period, which ensured a more accurate estimation of the seroprevalence. This is important as antibodies develop up to 19 days after having COVID-19 [10]. Another methodological strength is the fact that participants were not selected due to experiencing COVID-19 symptoms. One limitation of the study is the self-reported nature of the data. Participants had to perform the antibody tests themselves and report the results back to the study organizers. This may have caused misclassification of cases. However, if such bias existed it was likely to be random and therefore not affect the study results. Moreover, the relatively low sensitivity of the test (82.58%) potentially caused an underestimation of the seroprevalence. Testing participants multiple times may have reduced the level of underestimation. The potential underestimation was further reduced by additional testing of participants who tested positive for only one of the two SARS-CoV-2 antibodies, IgM or IgG. Another limitation of the study is the number of participants. Since the seroprevalence is low, the statistically insignificant findings may be an artefact of reduced statistical power. Moreover, because of the low seroprevalence and because the infection rate was low during the study period, we did not have the statistical power to investigate the development of the seroprevalence across time or to test for interactions. Descriptive statistics presenting the distribution of seropositivity among study participants seem to present a trend among Swedish employees showing that the more person contacts an employee had during a workday, the higher the proportion of employees with a positive test. We did not observe a similar trend among Danish participants. However, this is likely explained by the low number of Danish employees with more than 10 contacts per day. In line with this, another potential study limitation worth mentioning is the fact that we did not have information on other sources of COVID-19 transmission, such as social contacts in the employees’ spare time. However, we believe that the bias related to differences in social contacts would be random across job functions and therefore not impact the results significantly.

## 5. Conclusions

To conclude, this is the first bi-national investigation of SARS-CoV-2 seroprevalence among healthcare workers. Falck employs many people with different job functions, and therefore groups with different risks of becoming infected with SARS-CoV-2 were represented in this study. Findings represent an important contribution to surveillance of seropositivity in society and to the understanding of how this virus spreads. Such knowledge is imperative in constructing the most appropriate public health policies for dealing with the pandemic. Our study clearly shows that social interaction with customers or patients is the biggest risk factor for becoming infected with SARS-CoV-2 in our study population. Moreover, we observed a higher seroprevalence among Falck employees than among the background population in both countries, and we found a significant variance in seroprevalence between employees in Denmark and Sweden.

## Figures and Tables

**Table 1 ijerph-18-00923-t001:** Description of the cohorts in Sweden and Denmark. These descriptions represent the first test for each employee.

Characteristic		All Participants, *N* (%)	Sweden, *N* (%)	Denmark, *N* (%)	*p*
*N*		3272	1248	2024	
Sex					<0.001
	Men	1939 (59.3)	411 (32.9)	1528 (75.5)	
	Women	1333 (40.7)	837 (67.1)	496 (24.5)	
Age					<0.001
	<40	916 (28.0)	374 (30.0)	542 (26.8)	
	40–60	1732 (52.9)	687 (55.0)	1045 (51.6)	
	60+	624 (19.1)	187 (15.0)	437 (21.6)	
Employment					<0.001
	Ambulance staff	997 (30.5)	363 (29.1)	634 (31.3)	
	Firefighter	553 (16.9)	27 (2.2)	526 (26.0)	
	Healthcare staff	692 (21.1)	567 (45.4)	125 (6.2)	
	Office staff	717 (21.9)	267 (21.4)	450 (22.2)	
	Roadside assistance/field staff	313 (9.6)	24 (1.9)	289 (14.3)	
Customer or patient contacts/day					<0.001
	0	1043 (32.0)	254 (20.5)	789 (39.0)	
	1–5	1066 (32.7)	476 (38.4)	590 (29.2)	
	6–10	707 (21.7)	258 (20.8)	449 (22.2)	
	11–20	263 (8.1)	125 (10.1)	138 (6.8)	
	20+	185 (5.7)	127 (10.2)	58 (2.9)	
Hospital region					<0.001
	Norra sjukvårdsregionen	113 (3.5)	113 (9.1)		
	Uppsala-Örebro sjukvårdsregion	207 (6.3)	207 (16.6)		
	Stockholms sjukvårdsregion	490 (15.0)	490 (39.3)		
	Sydöstra sjukvårdsregionen	170 (5.2)	170 (13.6)		
	Västra sjukvårdsregionen	150 (4.6)	150 (12.0)		
	Södra sjukvårdsregionen	118 (3.6)	118 (9.5)		
	Region Hovedstaden	506 (15.5)		506 (25.1)	
	Region Sjælland	321 (9.8)		321 (15.9)	
	Region Syddanmark	349 (10.7)		349 (17.3)	
	Region Midtjylland	562 (17.2)		562 (27.9)	
	Region Nordjylland	277 (8.5)		277 (13.7)	
Former PCR test					<0.001
	Yes	1016 (31.5)	185 (14.8)	831 (42.1)	
	No	2207 (68.5)	1063 (85.2)	1144 (57.9)	

**Table 2 ijerph-18-00923-t002:** Number and proportion of employees with at least one positive test and its dependence on employee characteristics.

Characteristic		Number with a Positive Test Result (%)	Sweden Positive *N* (%)	Denmark Positive *N* (%)
Sex	Men	77 (4.0)	34 (8.4)	43 (2.8)
	Women	82 (6.2)	69 (8.3)	13 (2.6)
Age	<40	49 (5.4)	33 (9.0)	16 (3.0)
	40–60	90 (5.2)	60 (8.8)	30 (2.9)
	60+	20 (3.2)	10 (5.4)	10 (2.3)
Employment	Ambulance staff	79 (8.0)	53 (14.7)	26 (4.1)
	Firefighter	9 (1.6)	1 (3.8)	8 (1.5)
	Healthcare staff	33 (4.8)	31 (5.5)	<3
	Office staff	27 (3.8)	16 (6.1)	11 (2.5)
	Roadside assistance/field staff	11 (3.5)	<3	9 (3.1)
Customer or patient contacts/day	0	27 (2.6)	17 (6.7)	10 (1.3)
	1–5	50 (4.7)	28 (5.9)	22 (3.8)
	6–10	40 (5.7)	22 (8.5)	18 (4.0)
	11–20	20 (7.6)	16 (12.9)	4 (2.9)
	20+	21 (11.3)	19 (14.8)	<3
Hospital region	Northern healthcare region	Regions in Sweden	0	
	Uppsala-Örebro healthcare region		11 (5.4)	
	Stockholm Healthcare Region		62 (12.8)	
	Southeastern healthcare region		16 (9.5)	
	Western healthcare region		11 (7.4)	
	Southern healthcare region		3 (2.6)	
	Region Hovedstaden	Regions in Denmark		12 (2.4)
	Region Zealand			12 (3.8)
	South Denmark Region			9 (2.6)
	Central Jutland Region			18 (3.2)
	Region of northern Jutland			5 (1.8)

**Table 3 ijerph-18-00923-t003:** Logistic regression of risk of positive immune COVID-19 test.

	Model 1: Unadjusted		Model 2: Adjusted for Age and Sex		Model 3: Adjusted for Age, Sex and Region		Full Model *	
	OR (95% CI)	*p*	OR (95% CI)	*p*	OR (95% CI)	*p*	OR (95% CI)	*p*
Number of customer or patient contacts a day		0.00		0.00		0.00		0.02
0 (reference)	1		1		1		1	
1–5	1.8 (1.1–3.0)		1.9 (1.2–3.0)		1.6 (1.0–2.7)		1.4 (0.8–2.5)	
6–10	2.2 (1.4–3.7)		2.3 (1.4–3.9)		2.1 (1.3–3.5)		1.7 (0.9–3.0)	
11–20	3.1 (1.7–5.6)		3.3 (1.8–5.9)		2.6 (1.4–4.8)		2.3 (1.2–4.6)	
20+	4.8 (2.6–8.6)		4.6 (2.6–8.4)		3.4 (1.8–6.4)		2.9 (1.5–5.8)	
	Model 4:		Model 5:		Model 6:			
Employment type		0.00		0.00		0.00		0.06
Office staff/field staff (reference)	1		1		1		1	
Ambulance staff	2.2 (1.4–3.4)		2.7 (1.7–4.3)		2.1 (1.3–3.4)		1.4 (0.8–2.6)	
Firefighter	0.4 (0.2–0.9)		0.6 (0.3–1.3)		0.7 (0.3–1.6)		0.6 (0.2–1.3)	
Healthcare staff	1.3 (0.8–2.1)		1.2 (0.7–2.1)		1.0 (0.6–1.8)		0.8 (0.4–1.5)	
Roadside assistance	0.9 (0.5–1.9)		1.3 (0.6–2.8)		1.5 (0.7–3.2)		1.0 (0.5–2.3)	

Table 3 displays seven separate models, all with SARS-CoV-2 seropositivity as the dependent variable. Model 1: number of customer or patient contacts per day was included as the independent variable. Model 2: number of customer or patient contacts per day, age, and sex were included as the independent variables. Model 3: number of customer or patient contacts per day, age, sex, and region of residence were included as the independent variables. Model 4: employment type was included as the independent variable. Model 5: employment type, age, and sex were included as the independent variables. Model 6: employment type, age, sex, and region of residence were included as the independent variables. * The full model included the following covariates: sex, age, region, type of employment, and average number of customer or patient contacts during a workday.

**Table 4 ijerph-18-00923-t004:** Logistic regression of risk of positive immune COVID-19 test.

Only including Swedish Participants			Adjusted for Age and Sex		Adjusted for Age, Sex and Region in Sweden		Full Model *	
	Unadjusted							
	OR (95% CI)	*p*	OR (95% CI)	*p*	OR (95% CI)	*p*	OR (95% CI)	*p*
Number of customer or patient contacts a day		0.00		0.00		0.00		0.03
0 (reference)	1		1		1		1	
1–5	0.9 (0.5–1.6)		0.9 (0.5–1.7)		1.0 (0.5–1.9)		0.9 (0.4–1.8)	
6–10	1.3 (0.7–2.5)		1.3 (0.7–2.5)		1.4 (0.7–2.8)		1.2 (0.5–2.5)	
11–20	2.1 (1.0–4.2)		2.1 (1.0–4.3)		2.3 (1.1–4.7)		2.0 (0.9–4.5)	
20+	2.4 (1.2–4.8)		2.4 (1.2–4.8)		2.8 (1.4–5.7)		2.3 (1.0–5.1)	
Employment type		0.00		0.00		0.06		0.23
Office staff/field staff (reference)	1		1		1		1	
Ambulance staff	2.6 (1.5–4.7)		2.9 (1.6–5.2)		2.3 (1.2–4.3)		1.8 (0.9–3.7)	
Firefighter	0.6 (0.1–4.9)		0.7 (0.1–6.0)		0.7 (0.1–6.1)		0.4 (0.0–3.8)	
Healthcare staff	0.9 (0.5–1.7)		0.9 (0.5–1.7)		1.2 (0.6–2.2)		1.0 (0.5–2.1)	
Roadside assistance	1.4 (0.3–6.5)		1.7 (0.4–8.2)		1.2 (0.3–6.1)		1.1 (0.2–5.4)	

Table 4 displays seven separate models, all with SARS-CoV-2 seropositivity as the dependent variable. Model 1: number of customer or patient contacts per day was included as the independent variable. Model 2: number of customer or patient contacts per day, age, and sex were included as the independent variables. Model 3: number of customer or patient contacts per day, age, sex, and region of residence were included as the independent variables. Model 4: employment type was included as the independent variable. Model 5: employment type, age, and sex were included as the independent variables. Model 6: employment type, age, sex, and region of residence were included as the independent variables. * The full model included following covariates: sex, age, region, type of employment, and average number of customer or patient contacts during a workday.

## Data Availability

For access to data, please contact Jannie Laursen (jannie.Laursen@falck.dk). For access to applied computer code, please contact Janne Petersen (Janne.Petersen.01@regionh.dk).

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
