# Peer review of "Prevalence of SARS-CoV-2 IgG/IgM Antibodies among Danish and Swedish Falck Emergency and Non-Emergency Healthcare Workers"

_ijerph, 2021, doi:10.3390/ijerph18030923_

Round 1

Reviewer 1 Report

In a cohort study the authors investigated the prevalence of Covid antibodies in Swedish and Danish employees involved in health care activities.  Although the study has certain limitations which the authors clearers discuss the outcome is interesting and  helps to improve protective measures.

Some minor points need to be addressed for acceptance.

Results. There is no definition of the 4 rounds. Apparently the 4 different tests are meant. Explain under Methods.

It would be interesting to know whether and to what extent the positives had symptoms.

Some typos need to be corrected. E.g. in the Introduction people were masks instead of wear.

Author Response

Thank you for this helpful review. Please find our response to each comment below.

Reviewer 1:

  1. There is no definition of the 4 rounds. Apparently the 4 different tests are meant. Explain under Methods.

Authors response:             
We see why this can be confusing to the reader and we have now clarified in the Methods section that we have tested the participants 4 times every second week across a period of 2 months using the same test. Results from these four rounds of testing can be seen in the Results section on page 5 line 159-162: “After the first test 3.3% (n=107) was tested positive, after second test 4,1% (n=133) was tested positive, after three test 4,7% (n=153) and after fourth test 159 (4.9%) were tested positive for SARS-CoV-2 antibodies, corresponding to 2.8% of Danish participants and 8.3% of Swedish participants”

Changes in manuscript:    
The following was added to the Methods section on page 2, line 83: “Participants were tested for SARS-CoV-2 antibodies using the same test every second week for a period of 8 weeks from June 22, 2020 until August 10, 2020.

  1. It would be interesting to know whether and to what extent the positives had symptoms.

Authors response:             
We agree that it would be very relevant information to estimate the prevalence of asymptomatic SARS-CoV-2 positives. However, we are currently collecting data on this, which we plan to study in combination with a longer follow-up on late sequelae of COVID-19.

  1. Some typos need to be corrected. E.g. in the Introduction people were masks instead of wear.

Authors response:             
Thank you for noticing this error.                
We have now gone through the manuscript and believe that we have corrected all typos.

Reviewer 2 Report

Title:

“Prevalence of SARS-CoV-2 IgG/IgM antibodies among Danish 3 and Swedish Falck emergency and non-emergency healthcare workers”

Review:

-The manuscript is interesting. The objective of this review is to try to improve the manuscript. Some suggestions would be indicated:

  1. In a crude analysis, there is a significantly different in SARS-CoV-2 prevalence between Sweden and Denmark (Odds ratio [OR] = 3.18 95% 2.27-4.43) (1). On the other hand, this OR adjusted (age, sex, employment type and patient contacts a day) would be significant in accordance with the different actuations in Sweden and Denmark against COVID-19 pandemic.
  2. A dose-response p-value would calculate with a logistic regression model if number of customer or patient a day is included in the model, and not as an ordinal variable (0, 1-5, 6-10,11-20 20+).
  3. What is “Full model*”? It is not included in the Tables 3 and 4.
  4. In Tables 3 and 4, the ORs of a number of costumer or patient contacts a day and employment type appears to have been estimated joined. To avoid Table 2 fallacy (2), it would be better separate adjusting of the two variables.
  5. It would be convenient an indication of the participants who had COVID-19 symptoms before SARS-CoV-2 testing to know asymptomatic prevalence.
  6. Usually, relative risks are calculated in the cohort studies. Could you justify the use of OR in the study?
  7. As limitation of the study, other sources of COVID-19 transmission apart of job contacts, such as family or social contacts had not been considered.
  8. A consultation of the journal guideline may be recommended.

    Reference

1. 2x2 Contingency Tables with odds ratios, etc- VassarSats: Website for statistical computation. Available from http://vassarstats.net/odds2x2.html.

2. Westreich D, Greenland S. The table 2 fallacy: presenting and interpreting confounder and modifier coefficients. Am J Epidemiol. 2013;177:292-8.

Author Response

We thank the reviewer for these great comments. They were very helpful in improving the manuscript.

Reviewer 2:

  1. In a crude analysis, there is a significantly different in SARS-CoV-2 prevalence between Sweden and Denmark (Odds ratio [OR] = 3.18 95% 2.27-4.43) (1). On the other hand, this OR adjusted (age, sex, employment type and patient contacts a day) would be significant in accordance with the different actuations in Sweden and Denmark against COVID-19 pandemic.

Authors response:

We completely agree with the reviewer on the fact that national differences in actuations of COVID-19, which to some extent were affected by differences in national preventive policies would potentially have made for a significant difference in OR for getting infected with SARS-CoV-2 in Sweden versus Denmark. Therefore, we chose to report on the seroprevalence in both countries and instead of comparing the risk for getting infected between the countries, we compared the risk between different types of employments, and between reported number of social contacts during a workday. In the Discussion section we touched upon this issue when comparing SARS-CoV-2 prevalence between countries, on page 7, lines 202-209 where we wrote: It is plausible that the national variance in seroprevalence between the two countries is a result of different governmental strategies for dealing with the pandemic. The seroprevalences of 2.8% and 8.3% observed among Danish and Swedish Falck employees, respectively are higher than those observed among Danish (1.7%)[8] and Swedish (6.8%)[9] otherwise healthy blood donors. Blood donors represent an age and sex distribution similar to that of the background population between the ages 18 and 65. An explanation for part of the increase in the seroprevalence is that this expected to increase with time.” We hope that this answers the reviewer’s comment on this matter.

  1. A dose-response p-value would calculate with a logistic regression model if number of customer or patient a day is included in the model, and not as an ordinal variable (0, 1-5, 6-10,11-20 20+).

Authors response:             
This is a very relevant comment, which we appreciate- The reviewer is correct. This is not at formal dose-response trend. Therefore, we have rephrased this part in the manuscript.

Changes in manuscript:

A paragraph in the Results section on page 6 lines 172-174 was rephrased to the following: The multiple logistic regression analyses identified of being tested positive for SARS-CoV-2 antibodies with an increasing number of average people contacts during a workday.”

A paragraph in the Discussion section on page 7 lines 212-213 was rephrased to the following: The present results validate this, as analyses revealed an increasing risk of infection with an increasing number of customer or patient interactions.”

  1. What is “Full model*”? It is not included in the Tables 3 and 4.

Authors response:

Thank you for noticing this. We have now included a description of the full model in the footnotes of Table 3 and Table 4.

Changes in manuscript:

The following was added in the footnotes of Table 3 on page 6, line 187:  “*The full model included following covariates: sex, age, region, type of employment, and average number of social contacts during a workday”

The following was added in the footnotes of Table 4 on page 7, line 195:  “*The full model included following covariates: sex, age, region, type of employment, and average number of social contacts during a workday”

  1. In Tables 3 and 4, the ORs of a number of costumer or patient contacts a day and employment type appears to have been estimated joined. To avoid Table 2 fallacy (2), it would be better separate adjusting of the two variables.

Authors response:

We did estimate the impact of number of customer or patient contacts and employment type on risk of infection separately. Except in the last column of the tables where they also are mutually adjusted for each other. We have now clarified this by adding a more thorough description to the legends of Tables 3 and 4.

Changes in manuscript:

The following was added in the footnotes of Table 3 on page 6, line 187: 

“Table 3 displays seven separate models all with SARS-CoV-2 seropositivity as the dependent variable.

Model 1: Number of customer or patient contacts a day was included as the independent variable.

Model 2: Number of customer or patient contacts a day, age, and sex were included as the independent variables

Model 3: Number of customer or patient contacts a day, age, sex, and region of residence were included as the independent variables

Model 4: Employment type was included as the independent variable.

Model 5: Employment type, age, and sex were included as the independent variables

Model 6: Employment type, age, sex, and region of residence were included as the independent variables “

The following was added in the footnotes of Table 4 on page 7, line 195: 

“Table 4 displays seven separate models all with SARS-CoV-2 seropositivity as the dependent variable.

Model 1: Number of customer or patient contacts a day was included as the independent variable.

Model 2: Number of customer or patient contacts a day, age, and sex were included as the independent variables

Model 3: Number of customer or patient contacts a day, age, sex, and region of residence were included as the independent variables

Model 4: Employment type was included as the independent variable.

Model 5: Employment type, age, and sex were included as the independent variables

Model 6: Employment type, age, sex, and region of residence were included as the independent variables “

(The 7th model is the full model).

  1. It would be convenient an indication of the participants who had COVID-19 symptoms before SARS-CoV-2 testing to know asymptomatic prevalence.

Authors response:             
We agree that it would be very relevant information to estimate the prevalence of asymptomatic SARS-CoV-2 positives. However, we are currently collecting data on this, which we plan to study in combination with a longer follow-up on late sequelae of COVID-19.

  1. Usually, relative risks are calculated in the cohort studies. Could you justify the use of OR in the study?

Authors response:

This is true. However, an odds ratio can be interpreted similarly to a relative risk if the prevalence is relatively small which it is here. For the data collected in this study the logistic regression model fit data adequately, which is why we chose to fit such model.

  1. As limitation of the study, other sources of COVID-19 transmission apart of job contacts, such as family or social contacts had not been considered.

Authors response:

We appreciate this comment and we agree that it would have been highly relevant to include this information in the study. Unfortunately, we did not collect this information, but we have now included a short paragraph mentioning this in the Discussion part of the manuscript.

Changes in manuscript:

The following was added in the Discussion section on page 8, lines 261-265:

In line with this, another potential study limitation worth mentioning is the fact that we did not have information on other sources of COVID-19 transmission such as social contacts in the employees’ spare time. However, we believe that bias related to differences in social contacts would be random across job functions and therefore not impact the results significantly.

  1. A consultation of the journal guideline may be recommended.

Authors response:

We see that the journal editors have revised the format of the paper a bit before sending the review comments back to us. We hope that the manuscript is now in adherence with the journal’s guidelines. If not, we are happy to revise it accordingly.